# Unraveling the Fungal Community Associated with Leaf Spot on *Crataegus* sp.

**DOI:** 10.3390/microorganisms8030459

**Published:** 2020-03-24

**Authors:** Sonia Salazar-Cerezo, María de la Cruz Meneses-Sánchez, Rebeca D. Martínez-Contreras, Nancy Martínez-Montiel

**Affiliations:** 1Microbial and Molecular Ecology Laboratory, Research Center in Microbiological Sciences, Science Institute, Meritorious Autonomous University of Puebla, 72470 Puebla, Mexico; salcer_soni@hotmail.com (S.S.-C.); rebeca.martinez@correo.buap.mx (R.D.M.-C.); 2Department of Microbiology, Faculty of Chemical Sciences, Meritorious Autonomous University of Puebla, 72470 Puebla, Mexico; marie_qfb@yahoo.com.mx; 3Department of Microbiology and Infectiology, Faculty of Medicine and Health Sciences, University of Sherbrooke, Sherbrooke, QC J1E4K8, Canada

**Keywords:** *Crataegus*, fungi, leaf spot, phytopathogenic, ITS

## Abstract

*Crataegus* sp. is a tree that grows in temperate zones with worldwide distribution and is commonly known in Mexico as tejocote. The use of products derived from *Crataegus* in traditional medicine, food, and cosmetics has increased over the last few years and the relevance of this plant has also grown. Here, we report a disease that was observed in tejocote plants that grew both in the wild and in greenhouses in Puebla (Mexico). The disease was characterized by necrotic spots on the leaf ranging from brown to reddish tones that were accompanied by structures on the back of the leaf. Furthermore, we investigated the fungal genera associated with infected leaves in wild tejocote plants, from which we recovered *Alternaria* sp., *Aureobasidium* sp., *Dreschlera* sp., *Fusarium* sp., *Paecilomyces* sp. and *Ulocladium* sp. genera. Inoculation on healthy *Crataegus* sp. plants with isolate UAP140 showed similar symptoms as observed in nature, while inoculation with UAP127 resulted in the development of necrotic lesions in the leaf. The identity of these isolates was further studied through the phylogenetic analysis of the ribosomal DNA internal transcribed spacer (ITS) region, where isolate UAP140 showed the highest identity with *Fusarium equiseti* and isolate UAP127 was similar to *Alternaria arborescens*. To our knowledge, this is the first report of a characteristic disease developed in *Crataegus* sp. plants in Mexico where the fungal community associated to the lesion was analyzed. Further studies would be necessary to determine the ecological and environmental implications of the microbiome on the appearance and development of the disease.

## 1. Introduction

*Crataegus* sp. (Rosaceae: Maloideae), commonly known as hawthorn and in Mexico as tejocote, is an important cultivar with different uses in medicine [1] and the ornamental and food industries. Hawthorns grow as large shrubs or small trees that usually have thorns and produce bright to dark green leaves with margins that range from nearly entire to serrate or deeply lobed. The bushes or trees produce dense clusters of flowers and colored berries that vary between yellow and bright red or black [2]. *Crataegus* is native to northern temperate zones and it has been located in North America, East Asia, Central Asia and Europe, while in Latin America, it is mainly distributed in Mexico, Guatemala, Honduras, Costa Rica, Peru and Ecuador [3].

In Mexico, *Crataegus* is cultivated in the states of Mexico, Puebla, Tlaxcala, Chiapas, Michoacán, Hidalgo and Morelos [4], where the economic relevance of this cultivar is highly estimated. In medicine, it has been used to treat cardiovascular diseases [5,6], given that hawthorn fruits have been shown to have a tonic effect on the heart and are often used in the treatment of weak heart conditions, especially when accompanied by high blood pressure [7]. Hawthorn preparations have been used for their sedative actions to treat the early stages of congestive heart failure and to reduce total plasma cholesterol [8]. Its activity as an astringent allows its use in heavy menstrual bleeding and in diarrhea, while other properties such as antioxidant, antispasmodic, diuretic and antihypertensive functions have also been documented [7,9,10]. Traditionally, it has been used in the Mexican food industry to produce cakes, jams and beverages [11], but also as a source of pectin and vitamin C [12]. 

Unfortunately, microbial agents pose serious threats to *Crataegus* production around the world. Some fungal diseases have been reported for hawthorn species, including the cedar-hawthorn rust generated by *Gymnosporangium globosum*, the cedar-quince rust produced by *Gymnosporangium clavipes* [13] and the leaf blight and fruit rot associated with *Monilinia johnsonii*. Interestingly, only one report links this agent to *Crataegus* plants in Mexico [13] and there are no other reports for microbial agents with pathogenic effect in this species. Despite the cultural and economic relevance of this plant in Mexico, few efforts have been devoted to the identification of microbial pathogens that affect the health and production of this cultivar. During 2010–2011, leaf spot symptoms that evolved into necrotic lesions were observed on *Crataegus* leaves in several fields and greenhouses in the state of Puebla (Mexico). Initial lesions appeared as small, circular, dark brown spots. Subsequently, numerous lesions gradually enlarged with a necrotic center and brown margin until they occupied larger areas of the leaf. Similar lesions have been described in some *Crataegus* and other rosacea [13,14]; however, unlike for those reports, no lesions on the fruits developed (unlike the disease depicted here).

The objective of this work was the identification of the fungal agents that caused the characteristic spots accompanied by the appearance of structures on the back of tejocote leaves, a disease that was observed in *Crataegus* plants from Puebla (Mexico) and that we describe here for the first time. Given the economic relevance of *Crataegus* in Mexico, the development of improved strategies to mitigate the impact of this disease would be relevant to maintain this cultivar and its health.

## 2. Materials and Methods 

### 2.1. Collection of Leaves

Symptomatic and apparently healthy leaves were collected in the wild from the mountainous region called Tepeaca belonging to the state of Puebla, Mexico (latitude 98°07′37.62″, longitude 19°00′58.99″). *Crataegus* leaves were placed in sterile Petri dishes, placing each leaf in a different Petri dish, and transferred to the laboratory for further manipulation.

### 2.2. Fungal Isolation and Purification

Fungi were isolated from *Crataegus* leaves. Small sections of the diseased leaves were handled to recover the fungal population, while healthy leaves were also recovered from the same plant and processed in the same way. To identify the causative agent of the injury observed in the leaves, the structures developed on the surface of the leaf were cut and placed in potato dextrose agar (PDA) and V8 agar. To investigate if the isolates recovered from these structures were also occupying the plant as epiphytes, a wet cotton swab was used and passed over the leaf surface. The cotton was then immersed in 1 mL of sterile water, serial dilutions (10^−1^, 10^−2^ and 10^−3^) were prepared and 100 μL of each dilution were plated over PDA and V8 agar. Finally, to find out if the isolates were also inhabiting the plant as endophytes, fungal isolates were recovered from disinfected tissue. To accomplish this, the tissue surface was disinfected by immersion for 30 s in water, 1 min in 1% NaOCl and 3 min in 70% ethanol. After this treatment, the tissue was rinsed with sterile water, macerated, and serial dilutions (10^−1^, 10^−2^ and 10^−3^) were prepared. As before, 100 μL of each dilution was spread over PDA agar and V8 agar. All plates were incubated at 30 °C for 6 days. The isolates obtained were purified by re-inoculation in PDA or V8 plates.

### 2.3. Phenotypic Identification

General identification of the isolated fungi was performed using traditional morphological methods according to the macroscopic features (mycelium type, color and growth type) as well as the identification of the microscopic reproductive structures observed using the microculture technique [15], as depicted before [16]. The structures developed were compared to the reported references [17,18].

### 2.4. Inoculum Preparation

Isolates phenotypically correlated to the genera *Alternaria* sp. (isolates UAP035, UAP119 and UAP127), *Aureobasidium* sp. (isolates UAP078, UAP087 and UAP096), *Drechslera* sp. (UAP123), *Fusarium* sp. (UAP140, UAP168), *Paecilomyces* sp. (UAP133) and *Ulocladium* sp. (UAP108, UAP120) were individually grown for 4 days in yeast extract/phosphate/sucrose (YEPS) broth at 30 °C. Spore suspensions were obtained by vigorously mixing the inoculum. The thick suspension was then filtered. The concentration of the remaining suspension was counted with an hemocytometer and the suspension was diluted as necessary. Individual isolates were tested at a concentration of 8 × 10^6^ conidia/mL, while a mixture of individual isolates at the same concentration was also tested.

### 2.5. Plant Infection Assay

To evaluate the potential of the isolates to produce disease in the plant, 20 leaves of 3-month-old healthy plants of *Crataegus* were inoculated superficially or by injection with the inoculum prepared as depicted before. To inoculate the plant on the surface of the leaf, we created a 1-cm long laceration and then deposited 50 μL of the inoculum directly on the cut. In the inoculation through injection, we injected 100 μL of the inoculum directly on the midrib of the leaf to favor its distribution through the veins. As negative controls, 20 leaves were treated as described, but sterile media was used instead of the inoculum. Trees were monitored for 35 days after infection.

### 2.6. Molecular Identification and Phylogenetic Analysis

DNA was extracted from pure fungal cultures using the Wizard Genomic DNA Purification Kit (Promega, Madison, WI, USA) according to the manufacturer’s instructions. PCR amplification was carried out using the following primers: ITS1F (5′-CTTGGTCATTTAGAGGAAGTAA-3′) [19] and LR5 (5′-TCCTGAGGGAAACTTCG-3′) [20]. These primers were purchased from IDT Technologies Inc. (USA) and have been used previously to amplify the ITS region between the 18S and 28S rRNA fungal genes. PCR amplification was performed as previously described [21] in a MultiGene II Thermal Cycler (Labnet International, Inc., Edison, NJ, USA). The PCR protocol consisted of an initial denaturation step at 95 °C for 5 min, followed by 35 cycles of 35 s denaturation at 95 °C, 35 s annealing at 55 °C, and 2 min elongation at 72 °C. The PCR amplification was terminated by incubating for 10 min at 72 °C. Successful PCR amplification was confirmed by visualizing the products (7 μL) by electrophoresis on a 1% agarose/TBE gel and subsequent staining in ethidium bromide for 10 min. Single bands corresponding to different PCR products were purified from gel using the QIAEX II Gel Extraction Kit (QIAGEN, Germantown, MD, USA). Pure DNA was sequenced in both orientations using the Sequencing Services from MACROGEN (Korea) and compared to the sequences deposited in GenBank (NCBI). Sequence comparison and phylogenetic relationships were analyzed using the CLC software (QIAGEN, Germantown, MD, USA). The closest known relative for each sequence was determined using a BLAST(NCBI) search. The evolutionary relationship was inferred according to the Maximum Likelihood Phylogeny 1.2 method using CLC Main Workbench 7.9.1. The bootstrap analysis considered a total of 100 replicates.

## 3. Results

### 3.1. Disease Symptoms

During 2010–2011, lesions were observed in leaves of *Crataegus* trees that were growing in the wild and in greenhouses in several locations across Puebla (Mexico). In these plants, lesions appeared as small, circular, dark brown spots (Figure 1A,C). Besides, larger lesions with a necrotic center and brown margin occupying broader areas of the leaf were observed (Figure 1B). On top of this, lesions developed white structures on the back of the leaf accompanied by an almost total necrosis (Figure 1B,D).

### 3.2. Fungal Community Associated to the Leaf Spot in Crataegus spp Plants

To determine not only the fungal pathogen but also to assess the general fungal population of the *Crataegus* plants with this lesion, we intended to identify the fungal genus associated to the leaf spot. To accomplish this, fungal isolates were recovered from the structures, from the injured part of the leaf which constitutes the endophytic fungal population, and from the surface of the lesion that corresponded to the epiphytic fungal community. According to the phenotypical characteristics observed (Appendix A), 18 genera were identified inhabiting the diseased leaves of *Crataegus*.

Considering the origin of the sample, fungal incidence is presented in Table 1. A variable distribution as epiphytes, endophytes or isolates from the structures for the 18 genera was observed, with a total of 149 isolates identified. As expected, the largest population corresponded to the epiphytic community with 80 isolates, followed by the structures with 44 isolates and finally the endophytic group with 25 isolates. Overall, the most prevalent genera recovered from the three types of samples were *Aureobasidium* sp. and *Alternaria* sp. However, several genera also showed a significant distribution over the three different origins, including *Ulocladium* sp., *Paecilomyces* sp., *Dreschelera* sp. and *Fusarium* sp. Interestingly, some genera were recovered exclusively from one origin; that was the case for *Cephalosporium* sp., *Fonsecae* sp., *Phialophora* sp. and *Stemphylium* sp. (recovered only as epiphytes), while *Scopulariopsis* sp. and *Wangiella* sp. were found only as endophytes. The only genus restricted to the structures was *Trichosporum* sp. The general distribution of the different genera phenotypically identified is presented in Figure 2.

In a similar manner, fungal epiphytic and endophytic isolates were recovered from the tissue of healthy leaves from the same plant in order to analyze the fungal population inhabiting the plant from both healthy and diseased backgrounds. Using this approach, the genera recovered from healthy plants were *Aspergillus* sp., *Cephalosporium* sp. and *Penicillium* sp. Interestingly, *Aspergillus* sp. and *Penicillium* sp. were found only in healthy leaves and none of the genera found as endophytes or in the galls were identified in healthy tissue. *Cephalosporium* sp. was the only genus that was isolated from both healthy and diseased tissue, but in the last context, it appeared only once as an epiphyte (Figure 3).

Recapitulating, we recovered isolates associated to six genera that were highly prevalent in the injured leaves and that were recovered from the three different origins: as epiphytes, as endophytes and from the galls. These genera were *Aureobasidium* sp., *Alternaria* sp., *Dreschelera* sp., *Fusarium* sp., *Paecilomyces* sp. and *Ulocladium* sp.

### 3.3. Pathogenicity Assay

To assess the capability of the isolated fungi to produce an injury on healthy tejocote plants, some isolates that were recovered from the lesions of the tejocote leaves were inoculated individually and their effect on the leaf was evaluated. For this purpose, we inoculated the isolates indicated in Table 2 and plants were monitored for the appearance of signs of disease on the leaves.

We tested two different forms of inoculation: by injection or on the surface after laceration (as described in Material and Methods).

When inoculated by injection, isolates UAP140 and UAP168 showed the ability to develop a lesion in the leaves of healthy tejocote plants 10 days after the injection. Plants continued to be observed until day 35 after the injection, when the control plants remained asymptomatic (Figure 4) and none of the plants inoculated with the other isolates caused disease.

Plants inoculated with isolates UAP127, UAP140 and UAP168 developed spots on the leaf (Figure 4). However, while isolate UAP168 caused white spots on the leaf surface (Figure 4B), isolate UAP127 produced big necrotic lesions that extended on the surface of the leaf (Figure 4C). On the other hand, isolate UAP140 developed small brown spots surrounded by a yellow circle that highly resembled the features depicted for the original lesion observed in tejocote plants sampled at the beginning of this study (Figure 4A). On the other hand, leaves inoculated superficially did not develop any symptoms, even at the end of the observation period (Figure 4D).

### 3.4. Molecular Identification

To further characterize the isolates with a pathogenic effect in the plant, the molecular identification of the isolates UAP127 and UAP140 which produced discrete lesions on the surface of the leaves was done. To accomplish this, a pair of primers that have been designed to bind to a conserved region in the 18S and 28S genes and that allow the amplification of the ITS regions flanking the 5.8S gene were used. NCBI BLAST analysis showed that isolate UAP127 bears high similarity (99%) to the sequences of *Alternaria arborescens*, while isolate UAP140 was highly similar (99%) to *Fusarium equiseti*. The resulting sequences of the present isolates were submitted to GenBank (accession numbers MK742717 and MK680058, respectively).

The identified genera *Alternaria* and *Fusarium* are well known due to their ability to have a pathogenic effect over a broad spectrum of plants. However, given that the lesion developed in the pathogenicity test with isolate UAP140 was very similar to the original lesion observed at the beginning of this study, we continued our phylogenetic analysis using this isolate, phenotypically identified as *Fusarium* sp. To this end, isolate UAP140 was compared with reference sequences of *F. equiseti* and other *Fusarium* species while using three distant ascomycetes as an outgroup and the phylogenetic tree is presented in Figure 5. With these observations, our molecular results confirmed the identity of the isolates.

## 4. Discussion

### 4.1. Pathogens for Crataegus spp.

Traditionally, insects like *Rhagoletis pomonella* Walsh (Diptera: Tephritidae) are the most studied pests affecting *Crataegus* spp. orchards in Mexico [22]. However, microbial infections are starting to emerge in plantations and the detection of pathogenic agents is relevant in order to avoid damages in production. As leaf spot disease may pose a severe threat to commercial *Crataegus* producers, adequate pathogen identification and disease control is required to reduce economic losses.

According to the characteristics observed in the diseased tejocote plant, it was logical to propose that fungal strains could cause the observed lesion, mainly due to the structures that developed on the back of the leaf. So far, few reports of microbial pathogens for *Crataegus* species are available. Reported diseases for hawthorn include fire blight caused by *Erwinia amylovora*, the cedar-hawthorn rust generated by *Gymnosporangium globosum*, the cedar-quince rust produced by *Gymnosporangium clavipes* and leaf blight and fruit rot caused by *Monilinia johnsonii* [23]. *Diplocarpon mespili* (asexual: *Entomosporium mespili*) is also a common fungal pathogen causing leaf spot in *Crataegus* spp. [14]. However, none of the reported diseases correlate with the symptoms that are reported here.

### 4.2. The Fungal Community Associated to Diseased Crataegus spp. Plants

The detection of the fungal agent in plant material can be difficult, particularly when the pathogen is present at low infection levels or when the disease is advanced, providing a context where other pathogenic or opportunistic agents could increase the damage to the plant. In this study, a total of eighteen different genera were isolated from injured tejocote plants. Although some of the genera identified have been reported as phytopathogenic, there are no reports of a fungal genus associated to the lesion that we report here for tejocote plants.

In this study, the most prevalent genera were *Aureobasidium*, *Alternaria*, *Ulocladium*, *Paecilomyces*, *Drechslera* and *Fusarium*. However, the isolates with the ability to produce a disease in healthy tejocote plants corresponded to the genera *Alternaria* and *Fusarium*.

It has been reported that *Alternaria* sp. causes leaf spot for different hosts and in some cases, it can cause conidia to emerge from the injury (this was observed in this study for tejocote leaves) [24]. The genus *Alternaria* is ubiquitous and abundant in the atmosphere, soil, seeds and agricultural facilities. It includes plant pathogenic and saprophytic species that may affect crops in the field or can cause harvest and postharvest decay of plant products. [25] *Alternaria* sp. have been reported as a pathogen for apple, carnation, carrots, cruciferous, cucumber, tomato [26], geranium [27] and other plants, highlighting the need for a proper detection and control system. Some species have been associated to a particular disease. For example, *Alternaria iridiaustralis* was reported as the causative agent of leaf spot on *Iris ensata* in China [28]. Several efforts have been developed to identify plant pathogens in Korea, where *Alternaria tenuissima* was linked to the first report of leaf spot caused on black chokeberry (*Aronia melanocarpa*) [29] and *Alternaria simsimi* was associated to the first report of a pathogen causing leaf spot on sesame (*Sesamum indicum* L.) [30]. Another serious disease worldwide is leaf spot and blight disease of sunflower, which is caused by *Alternaria helianthi* (Hansford) Tubaki and Nishihara and molecular detection approaches have been developed in order to avoid economic losses [24]. In this regard, genotyping assays have been developed recently in order to identify *Alternaria* species causing the important and widespread citrus brown spot disease [31,32].

In this work, the isolate UAP140 that belongs to the genus *Fusarium* was able to produce similar lesions to those observed in diseased tejocote plants from Mexican plant orchards. *Fusarium* sp. has been reported as phytopathogen for beets [33], strawberry [34], corn [35] and lettuce [36], while different *Fusarium* species have been associated with the development of leaf spots in orchids [37], onion [38], and wild rocket [39], amongst others.

It has been reported that species of *Fusarium* and *Alternaria* could be pathogens of the same plant. For example, three fungal diseases of opium poppy in Eastern Uttar Pradesh have been reported [40], including damping of opium seedlings caused by *Fusarium solavi* (Mart) Sacc and leaf spot caused by *Alternaria alternata* (Fr) Keisler. According to the characteristics observed in the original lesion in correlation to the pathogenicity test performed in this work, it could be tempting to propose that the presence of both *Fusarium* and *Alternaria* species are necessary to produce the disease observed in tejocote leaves. These two fungal genera are widely distributed, and they could easily coexist in different environments. Interestingly, lesions similar to those reported here have been observed in other Rosaceae plants in Mexico and other locations, like Central Park in New York (personal observation; Rebeca Martínez and Nancy Martínez), suggesting that the disease is widespread in North America.

### 4.3. The Context of the Disease

Many of the species identified in this study have been previously reported as phytopathogens. However, none of them have been associated to the lesions reported in this work. On the other hand, some of the less frequently isolated species have usually been considered as opportunistic pathogens, like *Geotrichum* sp. that has only recently been associated with sour rot of loquat [41], fruit rot of strawberry [42] and sour rot of peaches in USA and Pakistan [43,44]. With the conditions tested in this study, we were not able to fully reproduce the original symptoms found in the diseased tejocote plants. For this reason, it would be tempting to propose that more than one fungal species could be responsible for the disease in the plant. Moreover, considering the abundance of fungal isolates recovered from diseased tejocote plants, we could also propose that there could be an existing condition necessary for the infection or for the appearance of the leaf spot with the structures on the back that we observed, given that (as for every organism) in plants, symptoms occur both because of the action of the pathogen and the response of the host.

Overall, this work constitutes the first report for characteristic leaf spot in *Crataegus* plants. We uncovered the fungal community associated to this lesion through phenotypic identification and we showed the capability of some of these isolates to have a pathogenic effect on healthy plants. Finally, we confirmed the identity of the isolates with phytopathogenic activity using a molecular approach.

To our knowledge, this study constitutes the first report aiming to identify pathogenic fungi causing leaf spot disease accompanied by leaf structures in Mexican tejocote orchards. However, further studies would be necessary in order to confirm that the isolate UAP127 that correlates with *Alternaria arborescens* and isolate UAP140 that proved to be similar to *Fusarium equiseti* are responsible for the characteristic lesion developed in *Crataegus* sp. and other Rosaceae.

## 5. Conclusions

In summary, this study provides a comprehensive view of the abundance, diversity, composition and pathogenic activity of the fungal population associated with *Crataegus* leaf spots, reported for the first time in this study. A combined phenotypic and molecular approach allowed the identification of 18 fungal genera associated with these diseased leaves, indicating that the general health condition of the plant is compromised due to the fungal infection. Furthermore, some isolates had the ability to develop different injuries in healthy tejocote plants. According to the phytopathogenic test, the isolate that was found to be highly similar to *Fusarium equiseti* was able to develop leaf spots similar to those depicted for the original lesion. The results from this work are relevant for further treatment of diseased tejocote plants, an important cultivar with different applications and great economic relevance to Mexican production.

## Figures and Tables

**Figure 1 microorganisms-08-00459-f001:**
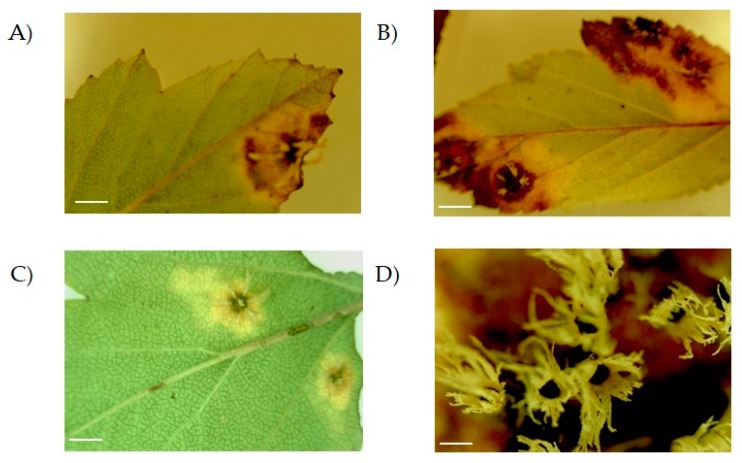
Lesions observed on leaves of *Crataegus* sp. plants. Samples were observed using the microscope with the 10X objective. The lesion evolved as a yellow and brown spot on the surface covering different parts of the leaf (compare (**A**), (**B**) and (**C**); scale bar = 0.5 cm). An enlarged view of the structures can be observed in (**D**) with the 40X objective; scale bar = 0.5 mm.

**Figure 2 microorganisms-08-00459-f002:**
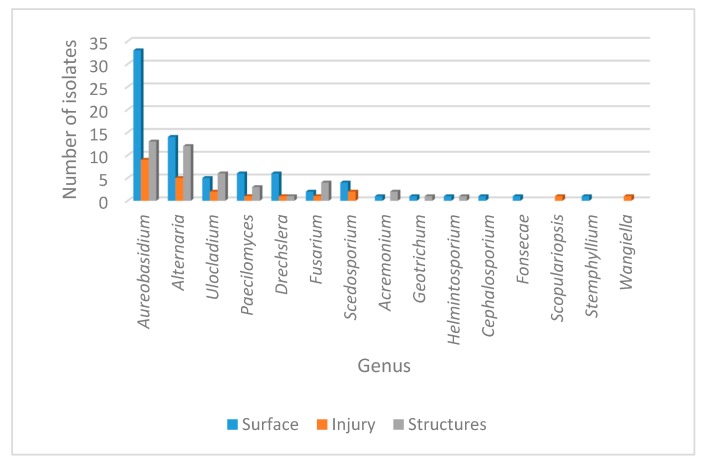
Abundance of fungal genera isolated from leaf disease in *Crataegus***.** The number of fungal isolates corresponding to the different genera identified phenotypically is shown. Isolate distribution is presented for the surface (blue), for injured tissue (orange) and for the structures (gray).

**Figure 3 microorganisms-08-00459-f003:**
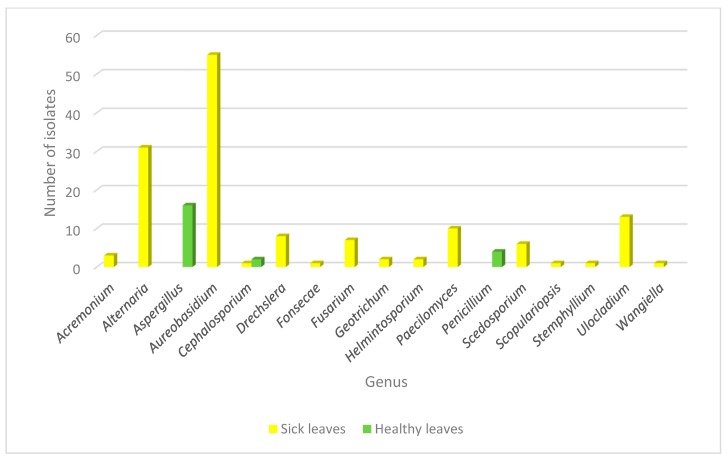
Abundance of fungal genera isolated from diseased and healthy leaves in *Crataegus*. Fungal distribution identified in the endophytic and epiphytic population in healthy (green) and sick (yellow) leaves is shown.

**Figure 4 microorganisms-08-00459-f004:**
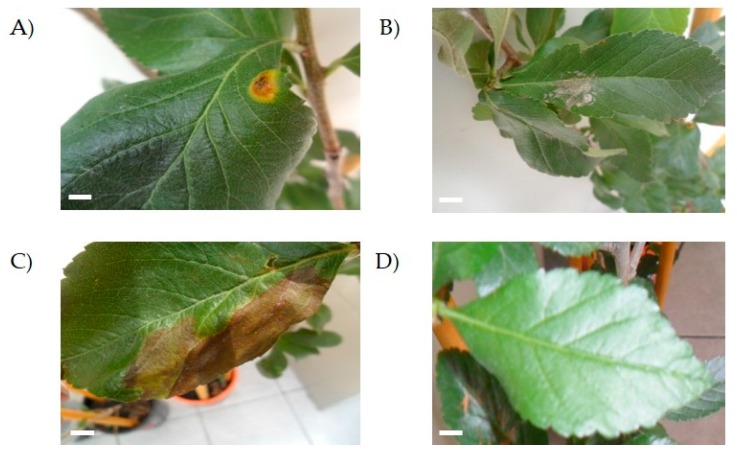
Pathogenicity test in healthy *Crataegus* sp. trees. The image shows injuries produced by isolates UAP140 (**A**), UAP168 (**B**) and UAP127 (**C**) in three-month-old trees. Control trees remained uninfected 35 days after infection (**D**). Scale bar = 0.5 mm.

**Figure 5 microorganisms-08-00459-f005:**
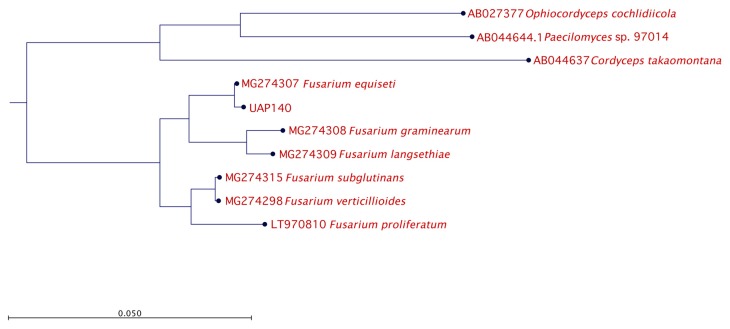
Phylogenetic relationship for isolate UAP140. The tree with the highest log likelihood is shown. The tree is drawn to scale, with branch lengths measured in the number of substitutions per site. The analysis involved six *Fusarium* reference nucleotide sequences with a total of 100 replicates in the bootstrap analysis, while three non-related fungal species were used as the outgroup.

**Table 1 microorganisms-08-00459-t001:** Frequency of isolation for the genera identified according to their origin.

Genus	Surface	Injury	Structures	Total
*Aureobasidium* sp.	32	9	13	54
*Alternaria* sp.	14	5	12	31
*Ulocladium* sp.	5	2	6	13
*Paecilomyces* sp.	6	1	3	10
*Drechslera* sp.	6	1	1	8
*Fusarium* sp.	2	1	4	7
*Scedosporium* sp.	4	2	0	6
*Acremonium* sp.	1	0	2	3
*Geotrichum* sp.	1	0	1	2
*Helmintosporium* sp.	1	0	1	2
*Cephalosporium* sp.	1	0	0	1
*Fonsecae* sp.	1	0	0	1
*Phialophora* sp.	1	0	0	1
*Scopulariopsis* sp.	0	1	0	1
*Stemphylium* sp.	1	0	0	1
*Wangiella* sp.	0	1	0	1
*Candida* sp.	4	2	0	6
*Trichosporum* sp.	0	0	1	1
**Total**	**80**	**25**	**44**	**149**

**Table 2 microorganisms-08-00459-t002:** Isolates evaluated in the pathogenesis assays and the genus phenotypically identified.

Genus	Isolate
*Alternaria* sp.	UAP035 UAP119 UAP127
*Aureobasidium* sp.	UAP078 UAP087 UAP096
*Fusarium* sp.	UAP118 UAP140 UAP168

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
