# Peer review of "Unraveling the Fungal Community Associated with Leaf Spot on Crataegus sp."

_microorganisms, 2020, doi:10.3390/microorganisms8030459_

Round 1

Reviewer 1 Report

ms microorganisms-710879 Unraveling the fungal community associated with 2 leaf spot on Crataegus sp.
by Salazar-Cerezo Sonia1, Meneses-Sánchez María de la Cruz2, Martínez-Contreras Rebeca D.1 and 4 Martínez-Montiel Nancy3
reviewer Herve Seligmann
This well written manuscript describes very professionally the identification of the fungal community of leaf lesions in tejocote.
I am not a mycologue and my molecular biology background is too limited to assess the validity of the methods.
Overall, descriptions are useful for diagnosis by others and show mastering of the subject. I commend the authors for the overview of the natural history of the host plant in the introduction, and their discussion of pathogen biology asd a potential community rather than single-species cause.
The manuscript should be published as it is (see minor typos I spotted and listed below).
line
178 phenotipically->phenotypically
268 Alteraria->Alternaria
references
double numbering, please correct

Author Response

We really appreciate the comments from the reviewer and the typos indicated were corrected.

Reviewer 2 Report

Review on the manuscript Salazar-Cerezo et al under the title ‘Unraveling the fungal community associated with 3 leaf spot on Crataegus sp.’

First of all, not everyone speak in Spanish and French and the names of the Universities have to be provided in English.

Line 27 ITS change to Internal transcribed spacer (ITS). You used the first time the name so you have to write the whole name of it.

The major mistake in the hole manuscript is the using of superscripts instead of the regular numbers to cite the articles. Please correct it, this is in the whole manuscript!

Lines 53-56 Any references?

Line 60-77 This is the results! You have to write about the current knowledge on the area of your research, without any description of the results!

Line 109 something wrong with the typing

Figure 1, 4 I missed the scale bar of the figure

Line 179 Remove one dot at the end of the sentence

Line 186 Figure with the capital letter

Line 208-210 You have in the publication Times New Roman font and it have to be different. Check journal requirements!

Line 224-225 different font again. Why didn’t you put the sequences in the manuscript supplementary material?

Line 231 Change 3 to three

Line 232 Change 5 to five

Figure 5 is nice, BUT I did not see any details how the authors received this tree? No detailed descriptions of the obtained phylogenetic tree!

The authors have to add the main conclusion into the manuscript.

To sum, the manuscript poorly prepared. The introduction is the weakest part of it and have to be significantly re-write. The phylogenetic analysis have to be explain in details in the MM section as well as in the results. The references part is not well prepared – double numbers everywhere, the latin names are without the italics, the names of the journals sometimes are represented as the short names and sometimes as the whole names. Any statistical analysis? (at least some basic ones!)

Author Response

We really appreciate the comments provided for the reviewer and we believe that the observations provided would allow us to enrich the quality of the manuscript. Here we present the detailed response to the observations.

First of all, not everyone speak in Spanish and French and the names of the Universities have to be provided in English.

The information is now provided in English

Line 27 ITS change to Internal transcribed spacer (ITS). You used the first time the name so you have to write the whole name of it.

The whole name was included as indicated.

The major mistake in the hole manuscript is the using of superscripts instead of the regular numbers to cite the articles. Please correct it, this is in the whole manuscript!

The correction was performed all along the manuscript.

Lines 53-56 Any references?

Some references were included according to the information available. However, there are really few reports of microbial infection for Crataegus in Mexico. This is stated now in the text.

Line 60-77 This is the results! You have to write about the current knowledge on the area of your research, without any description of the results!

Results were eliminated and the introduction was restructured and presented with additional information.

Line 109 something wrong with the typing

This sentence was corrected.

Figure 1, 4 I missed the scale bar of the figure

We do not include the scale bar, but we indicate in the figure legend that the image was acquired with the 40X objective.

Line 179 Remove one dot at the end of the sentence

The change was done.

Line 186 Figure with the capital letter

The correction was performed.

Line 208-210 You have in the publication Times New Roman font and it have to be different. Check journal requirements!

The requirements were fulfilled.

Line 224-225 different font again.

We corrected the font all along the manuscript

Why didn’t you put the sequences in the manuscript supplementary material?

The sequences are deposited in the genbank, according to the journal requirements.

Line 231 Change 3 to three

The change was performed.

Line 232 Change 5 to five

The change was performed.

Figure 5 is nice, BUT I did not see any details how the authors received this tree? No detailed descriptions of the obtained phylogenetic tree!

Additional information was included in the materials regarding the construction of the tree.

The authors have to add the main conclusion into the manuscript.

A conclusion is included in the revised version on the manuscript.

To sum, the manuscript poorly prepared. The introduction is the weakest part of it and have to be significantly re-write. The phylogenetic analysis have to be explain in details in the MM section as well as in the results. The references part is not well prepared – double numbers everywhere, the latin names are without the italics, the names of the journals sometimes are represented as the short names and sometimes as the whole names. Any statistical analysis? (at least some basic ones!)

We provided additional information regarding these observations and we hope to fulfill the requirements stated.

Reviewer 3 Report

The article titled "Unraveling the fungal community associated with leaf spot on Crataegus sp." can be published in microorganisms. But there are some issues in the manuscript. 

There are several English language mistakes; please correct them.

The introduction section is very weak and lacks citations and that too in a proper format.

In the M & M, section Inoculation preparation step is unclear.

In the pathogenicity assay in the result section is weak.

Molecular identification in the results is not clear to understand.

Conclusion section summarising the results must be added.

Author Response

We appreciate the comments from the reviewer and we present here the actions performed according to the comments.

There are several English language mistakes; please correct them.

We believe that we corrected all the mistakes.

The introduction section is very weak and lacks citations and that too in a proper format.

Introduction presents now additional information and citations, which are now presented in the proper format along the manuscript.

In the M & M, section Inoculation preparation step is unclear.

This information is now provided.

In the pathogenicity assay in the result section is weak.

We provided some additional information, hoping to clarify the section.

Molecular identification in the results is not clear to understand.

This part was re-written.

Conclusion section summarising the results must be added.

This section is now included

Round 2

Reviewer 2 Report

Dear Authors, 

the manuscript was significantly improved. I don't have any comments. 

best!

Author Response

We really appreciate all the time and effort devoted to the manuscript, and the suggestions that improved the quality of the work.

Thanks a lot!